# Comprehensive methodology for sample enrichment in EEG biomarker studies for Alzheimer's risk classification

Verónica Henao Isaza[1,2,3], David Aguillon[2], Carlos Andrés Tobón-Quintero[1], Francisco Lopera[2], John Fredy Ochoa-Gómez[2,3]*

1 Grupo Neuropsicología y Conducta (GRUNECO), Facultad de Medicina, Universidad de Antioquia (UdeA), Medellín, Colombia, 2 Grupo de Neurociencias de Antioquia (GNA), Facultad de Medicina, Universidad de Antioquia (UdeA), Medellín, Colombia, 3 Semillero de Neurociencias Computacionales (NeuroCo), Facultad de Medicina y Facultad de Ingeniería, Universidad de Antioquia (UdeA), Medellín, Colombia,

* john.ochoa@udea.edu.co

## Abstract

### Objective

Dementia, particularly Alzheimer's disease (AD), constitutes a major global health concern, with AD accounting for approximately 70% of all cases. EEG-based biomarkers hold promise for early identification of individuals at risk; however, small and heterogeneous samples frequently limit generalizability.

### Methods

An EEG-based sample enrichment framework was developed by integrating advanced signal processing, component-level feature extraction, data harmonization (neuroHarmonize), and Propensity Score Matching (PSM). EEG data from four independent cohorts were harmonized to reduce site-related variability while preserving covariates such as age and sex. Features including power, entropy, coherence, synchronization likelihood, and cross-frequency coupling were extracted from independent components. PSM was applied at 2:1, 5:1, and 10:1 ratios to expand and balance the control group (HC) relative to the Alzheimer's risk group (ACr), composed of PSEN1-E280A mutation carriers without cognitive symptoms.

### Results

Sample enrichment through PSM improved classification accuracy, with decision tree models yielding values between 0.91 and 0.96. Higher enrichment ratios enhanced model stability and generalizability, as shown by learning curves and confusion matrices. Feature selection was based on model performance and effect sizes (Cohen's d).

**Data availability statement:** "The publicly available datasets are: CHBMP: https://chb-mp-open.loris.ca/ SRM: https://openneuro.org/datasets/ds003775/versions/1.2.1 UdeA1 and UdeA2: https://openneuro.org/datasets/ds007427 The analysis codes are publicly available at: https://github.com/GRUNECO/eeg_harmonization https://github.com/GRUNECO/Data_analysis_ML_Harmonization_Proyect".

**Funding:** This work was supported by the Comité para el Desarrollo de la Investigación (CODI), Universidad de Antioquia (Project No. 2017-16371 to CATQ) and by the Comité para el Desarrollo de la Investigación (CODI), Universidad de Antioquia (Project No. PRG 2022-53407 to JFOG). The funders had no role in study design, data collection and analysis, decision to publish, or preparation of the manuscript.

**Competing interests:** The authors have declared that no competing interests exist.

## Conclusions

The proposed framework addresses sample size and variability constraints in EEG-based AD risk classification.

## Significance

Harmonization and statistical balancing provide a replicable strategy for multicenter EEG studies targeting early AD detection.

## Introduction

The treatment of neurodegenerative diseases, particularly Alzheimer's disease (AD), is emerging as a major global health concern [1]. AD, which is characterized by progressive cognitive decline, imposes a significant burden on individuals and society at large, contributing to approximately 70% of all cases of dementia worldwide [2]. Its hallmark features include the accumulation of beta-amyloid (Aβ) plaques and hyperphosphorylated tau neurofibrillary tangles, leading to neurodegeneration over time [3]. While understanding of the underlying mechanisms has advanced in recent decades, early and accurate detection of AD remains a major obstacle in clinical practice [4].

Several biomarkers for Alzheimer's disease have been identified in the literature; however, the use of electroencephalography (EEG) biomarkers for reliable Alzheimer's disease risk classification remains relatively unexplored [5]. Traditional methods often face challenges in balancing healthy non-carrier subjects (HC) and E280A mutation Alzheimer's disease carriers without clinically detectable cognitive impairment (ACr) groups. For this study, ACr refers to individuals who are asymptomatic carriers of the E280A mutation, defined as those without clinically detectable cognitive impairment according to neuropsychological assessments. Despite carrying the mutation, these carriers do not exhibit significant deficits in memory, language, or other cognitive functions, thus remaining cognitively intact at the time of assessment. Challenges also arise in accounting for demographic variables such as age and sex [6].

Despite the potential of EEG in distinguishing subjects in the preclinical stages of AD, obtaining sufficiently large samples for reliable comparisons remains a significant challenge. Recent studies have highlighted the importance of large-scale collaborations that emphasize the integration of diverse datasets [7–9]. The EEG-IP platform, as presented in the work of van Noordt et al. [10] serves as an exemplary model for successfully integrating infant EEG data from multiple sites. By pooling longitudinal cohort studies, adhering to the Brain Imaging Data Structure (BIDS) EEG standard, and implementing a common signal processing pipeline, a standardized and integrated dataset was achieved [11]. This pioneering effort highlights both the successes and challenges encountered, particularly in addressing issues related to signal annotation, timing, and independent component analysis [12] during preprocessing.

Similarly, the work of Duncan et al. [13] introduces the Data Archive for the BRAIN Initiative (DABI), a dedicated platform designed to address the complexities of sharing human intracranial neurophysiology recordings and multimodal data. This initiative aligns with the overarching goal of creating specialized repositories capable of accommodating the unique features of complex and heterogeneous datasets, further validating the feasibility and importance of harmonizing data from diverse sources [14].

The main objective of this study is to develop a framework for exploring differences in pre-symptomatic subjects with scarce samples using advanced machine learning (ML) techniques. Using propensity score matching (PSM) techniques [15], we aim to optimize the balance between HC and ACr. Our study provides a detailed analysis of the results obtained, allowing us to identify significant patterns and differences in the distribution of relative performance between the groups of interest. Data visualization through figures and tables helps us to better understand the peculiarities of the EEGs of ACr compared HC.

This paper presents original research focused on an innovative workflow that integrates data from multiple sources and employs advanced data analysis techniques. By addressing the urgent need to improve the early and accurate detection of Alzheimer's disease, we aim to provide new insights that will drive the development of more effective approaches to AD risk classification.

## Materials and methods

### Subjects and EEG acquisition

This study includes four EEG databases comprising healthy non-carriers (HC) and carriers of the PSEN1 E280A mutation associated with Alzheimer's disease (ACr), none of whom presented clinically detectable cognitive impairment at the time of recording. All EEG recordings were obtained under resting-state conditions.

### Inclusion and exclusion criteria

The selection of subjects and datasets followed predefined inclusion and exclusion criteria, summarized schematically in Fig 1. Briefly, included datasets were required to: (i) contain resting-state EEG recordings acquired with eyes closed and/or eyes open, (ii) include HC and/or ACr participants with available neuropsychological assessments, and (iii) be acquired using standard digital EEG systems with sufficient spatial resolution. Datasets were excluded if they were acquired using portable EEG systems, contained fewer than 58 electrodes, lacked detailed acquisition protocols, had been previously preprocessed, or corresponded to private sources without explicit consent from the data providers.

### EEG acquisition

Participants were seated comfortably in a quiet room and instructed to remain relaxed without performing any specific task. No external stimuli were presented during the recordings. Depending on the database, EEG was recorded with eyes closed and/or eyes open.

**General acquisition details across all databases:**

- **EEG systems:** Standard digital EEG systems were used, with 58–128 electrodes placed according to the 10−20 or extended 10−10 system.

- **Duration:** Resting-state recordings lasted between 4 and 10 minutes depending on the dataset.

- **Patient condition**: All participants were neurologically evaluated, with ACr participants showing no cognitive impairment. Basic demographic information, including age and sex, was collected for all participants (Table 1).

- **Data type:** Multi-center recordings included both neurophysiological (EEG) and neuropsychological information.

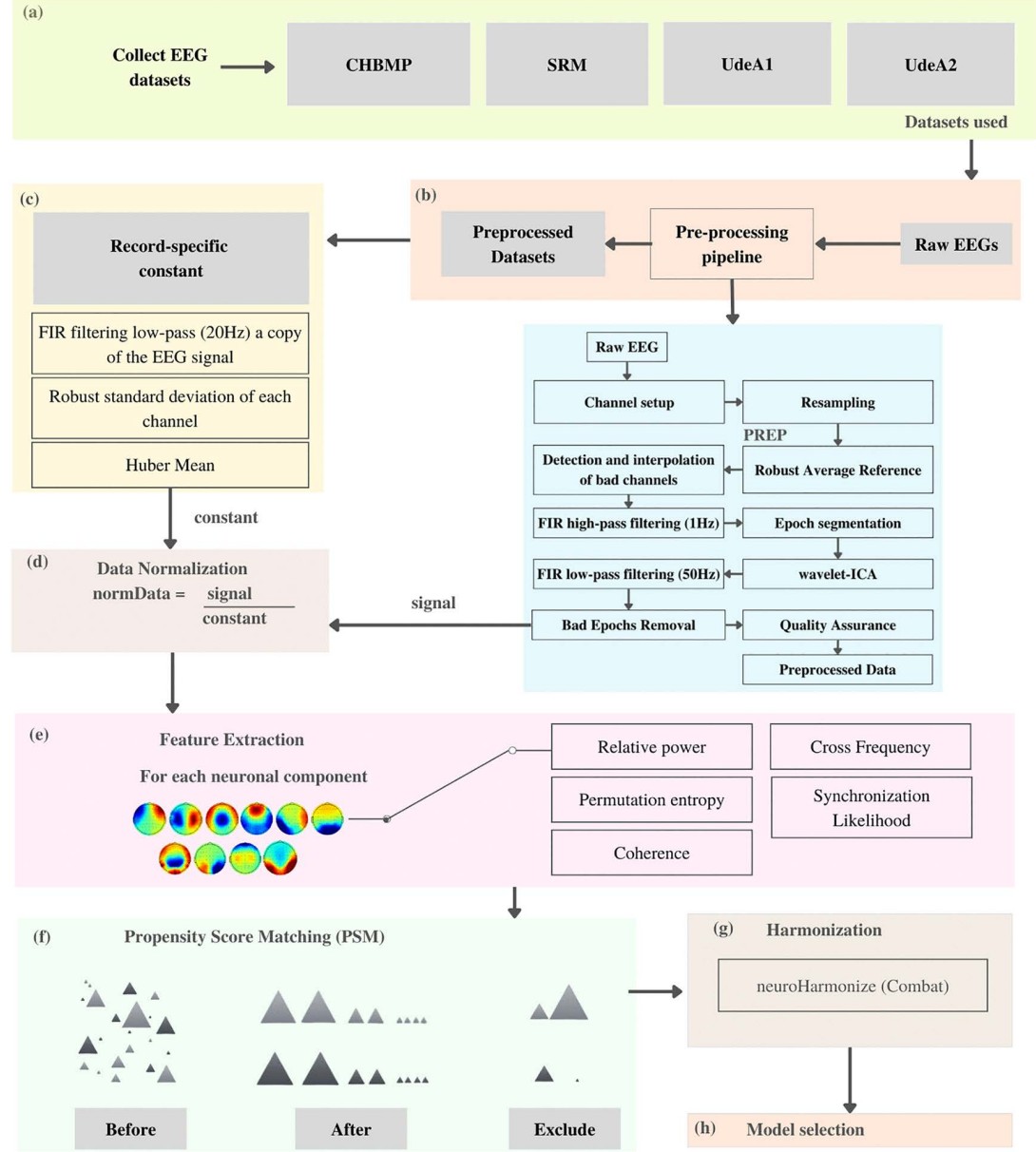

**Fig 1. EEG signal pre-processing and feature extraction pipeline.** Schematic representation of the standardized EEG pre-processing workflow applied in this study, including detrending, referencing, artifact removal using ICA and wavelet-ICA, normalization, feature extraction, and harmonization. The pipeline generates spectral, connectivity, and entropy-based features across multiple frequency bands and independent components.

### Database-specific details:

- **UdeA 1 Database:** 68 ACr, 77 HC. EEG was recorded for 5 minutes with eyes closed and open using a Neuroscan amplifier and a 58-channel tin cap.

- **UdeA 2 Database:** 11 ACr, 12 HC. EEG was recorded for 5 minutes with eyes closed using a Neuroscan amplifier and 64 electrodes.

**Table 1. Demographic and clinical characteristics of the EEG datasets.**

| | Database | Group | Count | MMSE | Education | Age | Sex (F/M) |
|---|---|---|---|---|---|---|---|
| 2:1 | CHBMP | HC | 38 | 29.34±1.23 | 12.39±1.69 | 27.63±6.67 | 13/25 |
| | SRM | HC | 31 | 29.48±0.92 | 11.87±3.08 | 30.77±5.21 | 19/12 |
| | UdeA1 | ACr | 68 | 29.40±0.87 | 10.25±3.37 | 35.81±4.36 | 49/19 |
| | UdeA1 | HC | 77 | 29.58±0.77 | 12.97±2.63 | 30.45±4.81 | 47/30 |
| | UdeA2 | ACr | 11 | 29.66±0.64 | 12.04±3.36 | 33.45±3.64 | 9/2 |
| | UdeA2 | HC | 12 | 29.40±1.07 | 11.70±4.35 | 31.42±7.15 | 10/2 |
| | Total | | 237 | | | | 147/90 |
| 5:1 | CHBMP | HC | 38 | 29.34±1.23 | 12.39±1.69 | 27.63±6.67 | 13/25 |
| | SRM | HC | 31 | 29.48±0.92 | 11.87±3.08 | 30.77±5.21 | 19/12 |
| | UdeA1 | ACr | 30 | 29.42±0.84 | 11.53±3.32 | 39.78±2.85 | 21/9 |
| | UdeA1 | HC | 77 | 29.58±0.77 | 12.97±2.63 | 30.45±4.81 | 47/30 |
| | UdeA2 | ACr | 1 | 28.00±0.00 | 11.00±0.00 | 43.0±nan | 1/0 |
| | UdeA2 | HC | 12 | 29.40±1.07 | 11.70±4.35 | 31.42±7.15 | 10/2 |
| | Total | | 189 | | | | 111/78 |
| 10:1 | CHBMP | HC | 38 | 29.34±1.23 | 12.39±1.69 | 27.63±6.67 | 13/25 |
| | SRM | HC | 31 | 29.48±0.92 | 11.87±3.08 | 30.77±5.21 | 19/12 |
| | UdeA1 | ACr | 14 | 29.00±1.07 | 11.50±2.98 | 41.86±2.35 | 12/2 |
| | UdeA1 | HC | 77 | 29.58±0.77 | 12.97±2.63 | 30.45±4.81 | 47/30 |
| | UdeA2 | ACr | 1 | 28.00±0.00 | 11.00±0.00 | 43.0±nan | 1/0 |
| | UdeA2 | HC | 12 | 29.40±1.07 | 11.70±4.35 | 31.42±7.15 | 10/2 |
| | Total | | 173 | | | | 102/71 |

HC: Healthy non-carrier subjects. ACr: E280A mutation Alzheimer's disease carrier without clinically detectable cognitive impairments.

- **SRM Database:** 31 HC. EEG was recorded for 4 minutes with eyes closed using a BioSemi ActiveTwo system with 64 electrodes following the extended 10−10 system.

- **CHBMP Database:** 38 HC. EEG was recorded for 10 minutes with eyes closed using a MEDICID digital system with 64 or 128 electrodes.

All datasets were pooled to form a comprehensive database including ACr and HC subjects. The main characteristics of each dataset are summarized in Table 1.

## Quality control

To ensure data reliability, quantitative quality-control metrics were applied at multiple stages of the EEG pre-processing pipeline. During the early-stage preprocessing (PREP pipeline), metrics were used to identify defective channels based on missing data (NaN), flat signals, amplitude deviation, high-frequency noise, correlation thresholds, signal-to-noise ratio, dropout events, and RANSAC-based detection. Artifact removal efficiency was further evaluated during wavelet–ICA processing by assessing the proportion of filtered independent components. Additionally, noisy time segments were identified and rejected using statistical and signal-based criteria, including kurtosis, amplitude thresholds, linear trends, and spectral power distribution.

The combined application of these metrics enabled systematic assessment of signal quality and ensured that only reliable EEG data were retained for subsequent analysis. A schematic summary of the quality-control metrics is provided in the Supplementary Materials (Supplementary S1 Fig).

## Ethics statement and data access dates

EEG data from the UdeA databases were collected prospectively under the approval of the Ethics Committee of the Instituto de Investigaciones Médicas at the Universidad de Antioquia (Approval Act No. 010, Code F-017–00). All participants provided written informed consent prior to participation. For the present study, access to the UdeA 1 and UdeA 2 datasets was carried out throughout the year 2022.

Data from the SRM and CHBMP databases are publicly available and were obtained under their respective terms of use. These datasets were collected by independent research teams with appropriate ethical approvals and were fully anonymized prior to release. Access to the SRM and CHBMP data for this study also took place throughout 2022. Therefore, no additional ethics approval or informed consent was required for their secondary use. The authors did not have access to any identifying participant information.

## EEG data pre-processing

Given the well-known inter-subject variability and stochastic nature of EEG signals, a standardized preprocessing, normalization, and harmonization strategy was applied to minimize non-neuronal variability while preserving biologically meaningful patterns.

The raw data underwent pre-processing using the pipeline proposed by Suarez et al. [16]. The standardized early-stage EEG (PREP) processing pipeline was applied (Fig 1), including signal detrending, robust referencing, and interpolation of bad channels. The Fast ICA algorithm obtained artifactual and neural ICA components after applying a high-pass filter. The records were segmented into 5-second epochs and subjected to wavelet-ICA for further artifact removal. At this step, individual Infomax ICA was applied. A low-pass filter was applied, and noisy epochs were detected and removed based on various criteria. The data was normalized to account for variability introduced by hair, scalp, and skull. Spectral, connectivity, and amplitude modulation features were extracted.

The Ochoa-Gómez, J. F., et al. [17] gICA methodology, outlined in the latter study, served as a robust foundation for extracting reproducible neuronal components from resting-state electroencephalographic data. This methodological framework ensures the reliability and reproducibility of the independent components, which were used as spatial filters for extracting the signals analyzed in this study.

Building on this, we applied machine learning models to harmonize features across diverse cohorts, focusing on Alzheimer's disease and PSEN1-E280A mutation carriers (ACr).

Additionally, feature extraction involved assessing several key metrics:

- **Relative Power:** This measure evaluates the proportion of a signal's power relative to a reference, providing insight into neural activity in different brain regions [18].

- **Permutation entropy:** A measure of uncertainty or disorder in a dataset, used to characterize the complexity and regularity of brain activity patterns [19].

- **Coherence:** A measure of the consistency or synchronization between signals at different frequencies, indicating functional communication between brain regions [20].

- **Cross Frequency:** Examines the relationship between oscillations in different frequency bands, providing insight into the organization and integration of neural activity [21].

- **Synchronization Likelihood:** Assesses the likelihood that two signals are synchronized in time, reflecting functional connectivity between brain regions [22].

## Harmonization

To ensure harmonization across different datasets, we employed the neuroHarmonize package [23]. This tool, which extends the functionality of neuroCombat [24], uses the ComBat algorithm for correcting multi-site data. The data matrix

and covariate matrix were prepared and harmonized, controlling for site effects while preserving covariate effects. This step was crucial for reducing inter-subject variability and improving the consistency of the data, thereby enhancing the reliability of subsequent analyses.

The propensity score matching (PSM) process used logistic regression to match individuals from the ACr and HC subjects based on their similarity in pretreatment covariates such as sex and age [15]. The propensity score calculated the probability of an individual being a gene carrier based on these characteristics. Subjects with lower propensity scores were then removed to achieve the desired proportion of subjects in the treatment group. PSM was used to ensure comparability between the healthy non-carrier (HC) and E280A mutation Alzheimer's disease carrier without clinically detectable cognitive impairment (ACr) groups in the study. This method aims to balance covariates between treatment and control groups in observational studies, thereby providing more valid comparisons and reducing bias due to non-random treatment allocation.

This logistic regression-based propensity score matching ensures that confounding variables are controlled, leading to more comparable and homogeneous groups. By refining the matching process, differences observed in the EEG data can be more accurately attributed to the gene carrier status rather than initial characteristic differences.

## Subject ratios and age-sex matching

We employed Propensity Score Matching (PSM) at varying ratios (2:1, 5:1, and 10:1) to assess the impact of sample size disparities between healthy non-carrier subjects (HC) and E280A mutation Alzheimer's disease carriers without clinically detectable cognitive impairment (ACr) on model performance. These ratios were chosen to explore how increasing the size of the HC cohort relative to ACr influences the robustness and generalizability of our findings. Through PSM, we aimed to achieve balance in demographic variables such as age and sex, thereby mitigating biases associated with non-random treatment assignment and improving the comparability between groups.

In Fig 2 the data input for both HC and ACr groups were combined across all cohorts (UdeA 1, UdeA 2, SRM, CHBMP) and subjected to propensity score matching (PSM). This process resulted in matched datasets with two, five, and ten times as many HC as ACr.

## Model selection

The data was organized within dataframes, where each row corresponded to a record, and the columns represented specific features. For the implementation and validation of the model, an 80−20 train-test split was applied, ensuring that class proportions were maintained to provide an unbiased model evaluation on unseen data.

The cross-validation process employed a pre-fitted estimator for predicting outcomes, using ten-fold iterations for training and evaluation. Performance scores were averaged across folds to ensure robust model assessment.

The pseudocode (Fig 3) outlines the feature selection and model evaluation process, which highlights the steps taken to identify the most relevant features and optimize model precision:

Following this process, the features that contributed most to improving model accuracy were selected. The decision tree model, which achieved the highest precision, was subsequently employed to generate a confusion matrix, effectively evaluating the model's ability to classify individuals at risk of Alzheimer's disease.

Decision trees were chosen for their intrinsic feature selection capabilities, enabling a deeper analysis of feature importance. This process allowed us to assess how individual features influenced the classification of patients, providing insights into both their individual and collective impact on model performance.

Additionally, the effect size of each selected feature was evaluated using Cohen's d. This standardized measure quantified the difference between ACr and HC groups for each specific metric, offering a comprehensive understanding of each feature's contribution to group differentiation.

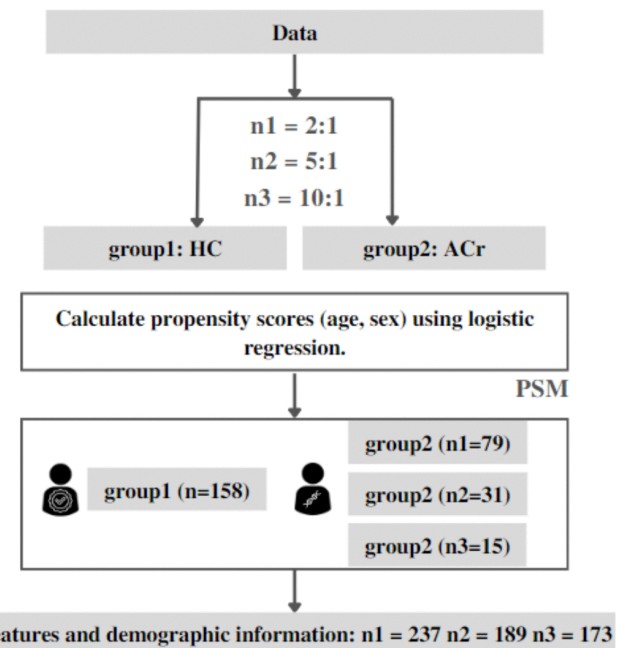

**Fig 2. Propensity score matching across multicenter EEG datasets.** Age- and sex-matched records from healthy non-carrier subjects (HC) and E280A mutation Alzheimer's disease carriers without clinically detectable cognitive impairment (ACr) were combined across four cohorts and matched using propensity score matching at 2:1, 5:1, and 10:1 ratios.

## Results and discussion

Our study utilized an extensive dataset to develop a model capable of identifying statistical differences and distinguishing between ACr and HC across different cohorts. This comprehensive approach allowed us to provide a more nuanced and precise understanding of the implications of the PSEN1-E280A mutation in varied carrier contexts. The findings have significant clinical and research implications, offering potential advancements in early diagnosis and targeted interventions for Alzheimer's disease. By integrating data from diverse sources, our model enhances the robustness and generalizability of EEG-based classification, contributing valuable insights to the field of Alzheimer's research.

### Preprocessing

By implementing the previously validated processing pipeline from the study of Henao Isaza et al. [25]. The relative power spectral density was computed by analyzing EEG frequency bands, including delta (1.5–6 Hz), theta (6–8.5 Hz), alpha 1 (8.5–10.5 Hz), alpha 2 (10.5–12.5 Hz), beta 1 (12.5–18.5 Hz), beta 2 (18.5–21 Hz), beta 3 (21–30 Hz), and gamma (30–45 Hz) [26].

In Fig 4, The graph illustrates the comparison of components (ICs) of the Alpha2 band across four cohorts of interest. Each boxplot represents the distribution of relative power values for a specific component (C) within different groups and databases.

For component C1, the median power values range from 0.05 to 0.1 across all subjects in the four cohorts. However, outliers are observed for the SRM and CHBMP HC groups, with the CHBMP group exhibiting an outlier reaching a value of 0.30 for relative power. In addition, only the UdeA1 cohort shows an outlier for the ACr group.

For Component C2, the median power values across subjects are even closer, with outliers detected in the CHBMP and UdeA2 cohorts. Notably, in the UdeA2 cohort, outliers are present in both the HC and ACr groups. Component C3

**Feature Selection and Model Evaluation Process**

**Input:** Harmonized data D, feature set F, accuracy threshold T
**Output:** Final model M, confusion matrix C

1: Load harmonized data: D = load_data()
2: Initialize selected feature list: S = []
3: for each feature f in F do
4:      Train model with current feature: M_f = train_model(D, f)
5:      Evaluate model accuracy: A_f = evaluate_model(M_f)
6:      if A_f ≥ T then
7:          Add feature to selected list: S.append(f)
8:      end if
9: end for
10: Assign weights to selected features: W = assign_weights(S)
11: Retrain model with selected features: M = train_model(D, S, W)
12: Generate confusion matrix: C = generate_confusion_matrix(M)
13: Return final model M and confusion matrix C

**Fig 3. Feature selection and model evaluation workflow.** Diagram illustrating the iterative feature selection process applied to harmonized EEG data, including correlation-based feature reduction, decision tree-based importance ranking, model training, and performance evaluation using confusion matrices.

shows greater variability in median power values, with outliers in the UdeA2 cohort. Components C4, C5, C6 and C8 show a higher prevalence of outliers, especially in the HC group. In components C7 and C9, the median power values range from 0.075 to 0.150, with outliers observed in the SRM, UdeA1, and UdeA2 cohorts for the ACr group.

## Propensity score matching (PSM)

By estimating the propensity score, which is the probability of receiving a particular treatment given observed covariates, individuals in the treatment group can be matched with individuals in the control group who have similar propensity scores.

As shown in Fig 5, the range of common support by treatment status. The propensity score is plotted on the x-axis and the frequency is plotted on the y-axis. The common support functions are smooth, and the algorithm's balancing property is satisfied. Beneficiaries outside the range of shared support were dropped in the logistic regression models for each case (2:1, 5:1, 10:1).

## Feature selection

It is important to select relevant features that capture meaningful aspects of brain activity before training a model for EEG analysis. The features used are presented in Fig 1, including the components from [17] and the frequency bands described in the processing, some of which are shown in Fig 6. These features were selected based on their relevance to capturing meaningful aspects of brain activity for EEG analysis. From nearly 967 initial features (Fig 1), the model first removed those with the highest correlations. The decision tree algorithm then identified the 100 most important features for inclusion (Model Selection). Cohen's d values were utilized to quantify the magnitude of differences between groups ACr and HC for each specific metric.

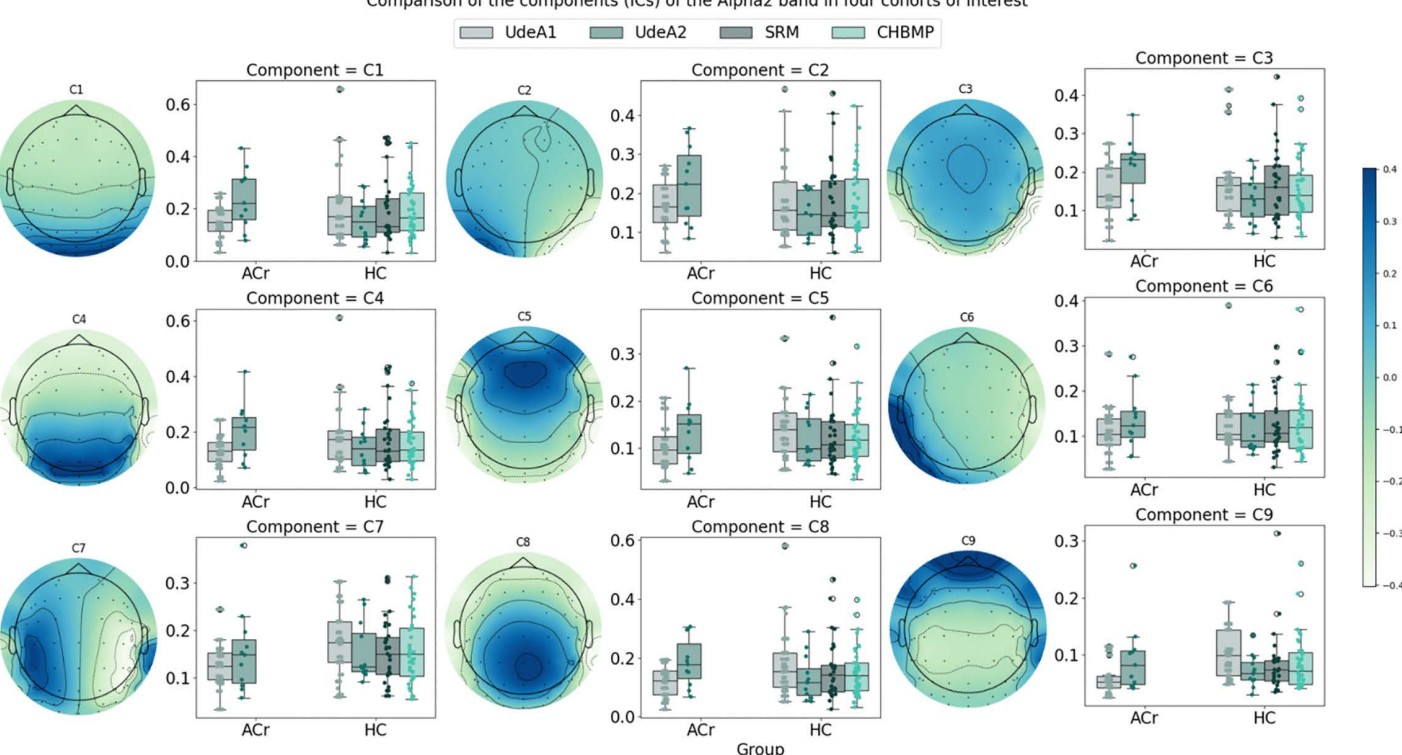

**Fig 4. Relative power in the Alpha 2 band across cohorts.** Boxplots showing the distribution of relative power values for independent components in the Alpha 2 frequency band across four cohorts, separated by healthy non-carrier subjects (HC) and asymptomatic E280A mutation carriers (ACr).

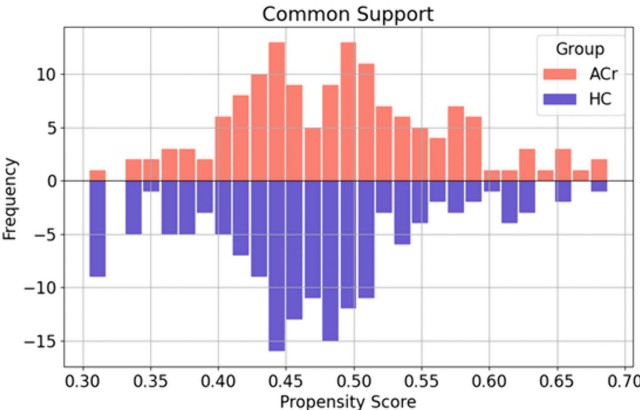

**Fig 5. Area of common support after propensity score matching.** Distribution of propensity scores for healthy non-carrier subjects (HC) and E280A mutation Alzheimer's disease carriers without clinically detectable cognitive impairment (ACr), illustrating the region of common support used for matching.

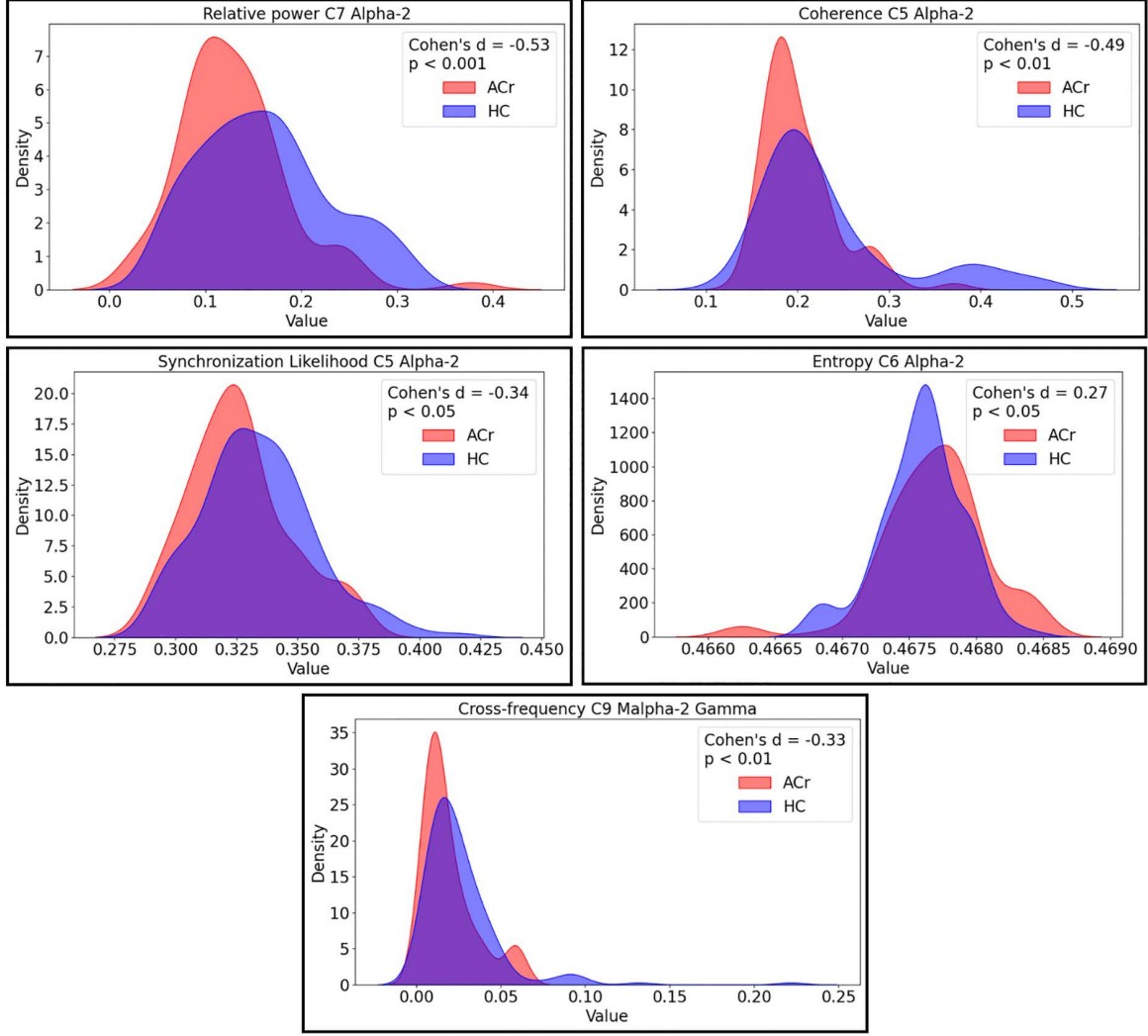

**Fig 6. Distribution of EEG feature effect sizes.** Kernel density plots illustrating the distributions of Cohen's d for selected EEG features, including spectral power, functional connectivity, synchronization likelihood, entropy, and cross-frequency coupling measures. Positive or negative values indicate the direction of differences between healthy non-carrier subjects (HC) and asymptomatic E280A mutation carriers (ACr). Statistical significance was assessed using the Mann–Whitney U test.

Fig 6 illustrates the distribution of effect sizes (Cohen's d) for selected metrics in EEG analysis, highlighting the observed differences between ACr and HC groups. Cohen's d provides standardized measures of effect size: Small (0.2): Indicates a small difference. Medium (0.5): Represents a moderate difference. Large (0.8): Indicates a substantial difference. Very Large (1.2+): Represents a significant difference.

Relative power analysis revealed a Cohen's d of −0.53 for component C7 in the Alpha2 band, indicating a moderate-to-large effect size and a highly significant difference (p < 0.001) between the ACr and HC groups. Coherence analysis for component C5 in the Alpha2 band showed a Cohen's d of −0.49, corresponding to a moderate effect size and a highly significant difference (p < 0.01). Synchronization likelihood in component C5 within the Alpha2 band yielded a Cohen's d of −0.34, suggesting a small-to-moderate effect. Entropy analysis for component C6 in the Alpha2 band

resulted in a Cohen's d of −0.27, indicating a small effect size. Finally, cross-frequency coupling analysis in component C9 showed comparatively smaller effects.

In addition, these features were evaluated using Cohen's d to assess effect size, comparing group differences across three proportions (n1, n2, and n3). An initial model training was conducted on the n1 dataset, followed by incremental model adjustments on n2 and n3 datasets. This iterative approach ensured comprehensive model refinement and evaluation across different dataset compositions.

The analysis of key EEG features across the different sample ratios (Table 2) revealed even more pronounced differences between groups. In the 2:1 ratio, relative power in components C7 and C10 within the Beta3 band showed very large effect sizes (Cohen's d = 1.22 and 1.12, respectively; p < 0.0001). As the augmentation ratio increased, the discriminative power of cross-frequency coupling features became dominant. Specifically, in the 5:1 ratio, interactions such as C8 Mbeta3-Theta and C9 Malpha1-Theta exhibited large negative effects (Cohen's d = −1.69; p < 0.0001). This trend culminated in the 10:1 ratio, where features like C3 Mbeta3-Alpha1 and C5 Mbeta2-Alpha2 reached extreme effect sizes (Cohen's d = −2.28; p < 0.0001). Notably, age remained a highly significant covariate across all ratios, with its effect size increasing from 1.10 in the 2:1 scenario to 2.78 in the 10:1 scenario, underscoring the importance of demographic harmonization in the classification models.

## Model performance across ratios

For the 2:1 ratio, the training curve starts at around 0.5 scores and gradually increases, stabilizing at approximately 0.8 after 30 samples. In contrast, the validation curve begins around 0.4 scores, exhibits a slightly steeper rise, stabilizes at about 0.83 after 30 samples, and then climbs to about 0.87 after 130 samples, peaking finally at approximately 0.95.

In the case of the 5:1 ratio, the training curve starts near a score of 0.99, dips slightly around 50 samples, and rises again to about 1.0 by 100 samples. Meanwhile, the validation curve begins at a score of 0.83, experiences a slight dip, rises again around 40 samples, and gradually increases to about 0.97.

**Table 2. Key EEG features differentiating ACr and HC subjects across sample ratios (Cohen's d and Bonferroni p-values).**

| | Feature | Cohen's d | p bonferroni |
|---|---|---|---|
| 2:1 | Relative power C7 Beta3 | 1.22 | <0.0001 |
| | Relative power C10 Beta3 | 1.12 | <0.0001 |
| | Age | 1.10 | <0.0001 |
| | Relative power C9 Alpha1 | −1.05 | <0.0001 |
| | Cross Frequency C8 Mbeta3 Gamma | 1.00 | <0.0001 |
| | … | | |
| 5:1 | Age | 2.22 | <0.0001 |
| | Cross Frequency C8 Mbeta3 Theta | −1.69 | <0.0001 |
| | Cross Frequency C8 Mgamma Theta | −1.69 | <0.0001 |
| | Cross Frequency C9 Malpha-1 Theta | −1.69 | <0.0001 |
| | Cross Frequency C9 Malpha-2 Theta | −1.69 | <0.0001 |
| | … | | |
| 10:1 | Age | 2.78 | <0.0001 |
| | Cross Frequency C3 Mbeta1 Alpha-1 | −2.28 | <0.0001 |
| | Cross Frequency C3 Mbeta2 Alpha-1 | −2.28 | <0.0001 |
| | Cross Frequency C3 Mbeta3 Alpha-1 | −2.28 | <0.0001 |
| | Cross Frequency C5 Mbeta2 Alpha-2 | −2.28 | <0.0001 |
| | … | | |

Lastly, for the 10:1 ratio, the training curve starts near a score of 1.00 and maintains a near-perfect trajectory through-out. The validation curve for this ratio begins at approximately 0.9 and grows steadily until it converges at 1.00, demonstrating the most stable and optimal performance among all ratios, consistent with the perfect accuracy and precision scores reported in Table 3.

The learning curves presented in Fig 7 complement the statistical results from Table 3 by illustrating the model's convergence and stability across different augmentation ratios. For the 2:1 ratio, a noticeable gap between training and validation accuracy persists in the early stages, suggesting that smaller samples limit the model's initial generalizability. However, as the ratio increases to 5:1 and 10:1, this gap narrows significantly and earlier in the process. Specifically, the 10:1 ratio exhibits the fastest convergence, with the validation curve reaching a plateau of 1.00 almost simultaneously with the training curve.

## Confusion matrix

In the following Fig 8 three confusion matrices for class distribution ratios of 2–1, 5–1, and 10–1, providing a visual representation of the computer precision results. In these matrices, TP stands for true positives, FP for false positives, FN for false negatives, and TN for true negatives.

For the 2:1 ratio, the results are TP = 31, FP = 1, FN = 3, and TN = 13. For the 5:1 ratio, the results are TP = 32, FP = 0, FN = 1, and TN = 5. For the 10:1 ratio, the results are TP = 32, FP = 0, FN = 2, and TN = 1. These matrices provide insight into the performance of the model at different class distribution ratios, highlighting the differences in correct and incorrect predictions.

Integrating data from multiple cohorts is fundamental to improving the robustness and generalizability of EEG-based AD detection models [27]. Our study took a comprehensive approach by combining data from multiple sources. This highlights the importance of considering the diversity of data in Alzheimer's disease research to ensure that findings apply to a wide range of populations and clinical scenarios [28].

**Table 3. Classification performance metrics across matching ratios.**

|  | Metric | Mean (10 folds) ±SD | CI 95% lower | CI 95% upper |
|---|---|---|---|---|
| 2:1 | Accuracy | 0.95 ± 0.04 | 0.92 | 0.97 |
|  | AUC | 0.98 ± 0.03 | 0.96 | 0.99 |
|  | Recall | 0.89 ± 0.13 | 0.81 | 0.97 |
|  | Precision | 0.96 ± 0.06 | 0.92 | 1.00 |
|  |  |  |  |  |
| 5:1 | Accuracy | 0.97 ± 0.04 | 0.96 | 1.00 |
|  | AUC | 0.99 ± 0.00 | 1.00 | 1.00 |
|  | Recall | 0.97 ± 0.24 | 0.74 | 1.00 |
|  | Precision | 1.00 ± 0.00 | 1.00 | 1.00 |
|  |  |  |  |  |
| 10:1 | Accuracy | 1.00 ± 0.00 | 1.00 | 1.00 |
|  | AUC | 1.00 ± 0.00 | 1.00 | 1.00 |
|  | Recall | 1.00 ± 0.00 | 1.00 | 1.00 |
|  | Precision | 1.00 ± 0.00 | 1.00 | 1.00 |
|  |  |  |  |  |

Table 3 presents the results of the computer precision function, which likely calculates the precision (and possibly other metrics) on the test dataset. By comparing the model's predictions with the true labels, this result is crucial as it evaluates the model's performance on unseen data, providing a more realistic estimate of how the model will perform on future data.

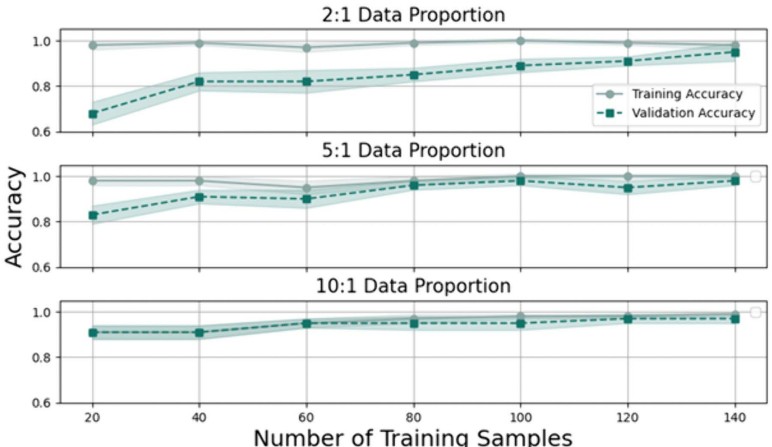

**Fig 7. Learning curves for different class distribution ratios.** Training and validation accuracy curves for decision tree models trained using 2:1, 5:1, and 10:1 ratios of healthy non-carrier subjects (HC) to asymptomatic E280A mutation carriers (ACr).

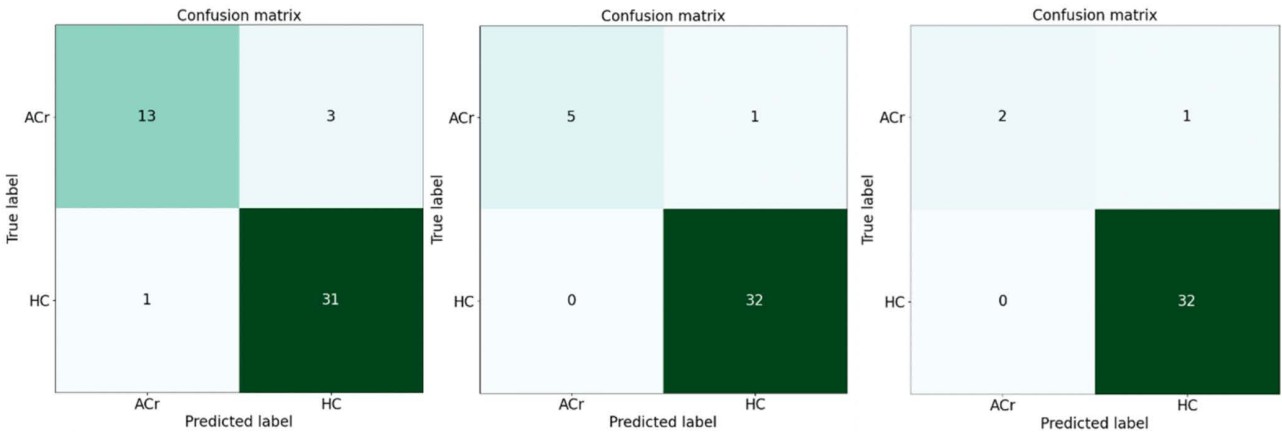

**Fig 8. Confusion matrices for classification performance.** Confusion matrices obtained from the test datasets for decision tree models trained using 2:1, 5:1, and 10:1 HC-to-ACr ratios. Performance was evaluated on 20% of the total data.

Our discussion focuses on how PSM and gICA improve the accuracy and reliability of our model, highlighting the importance of considering different methodological approaches to address unique challenges in early AD detection. This comparison underscores the need for comprehensive and multifaceted strategies to improve the accuracy and applicability of EEG-based AD detection models [29].

The figure in Fig 7 illustrate the impact of different class distribution ratios on model stability. For the 2:1 ratio, both curves show a gradual increase, with the validation accuracy reaching 95%, matching the results in Table 3. For the 5:1 ratio, the validation curve exhibits a slight initial dip before climbing to 97%. Notably, at the 10:1 ratio, the model demonstrates its most robust performance; both curves quickly converge and reach a perfect accuracy of 100%, showing the greatest stability and steady growth over time.

The performance metrics in Table 3 further support these observations. For the 2:1 ratio, the model achieved an accuracy of 95% and an AUC of 98%. As the sample enrichment increased, performance improved: the 5:1 ratio reached an accuracy of 97% and a perfect precision of 1.00, although with a slightly lower recall of 0.97. The 10:1 ratio yielded the most exceptional results, with accuracy, AUC, recall, and precision all reaching 1.00.

This perfect performance in the 10:1 configuration suggests that a higher ratio of healthy controls to carriers facilitates an optimal decision boundary for the algorithm. This setup likely provides a superior equilibrium after harmonization and matching, allowing the model to discriminate between classes with no error. These findings are particularly significant as they exceed recently reported AUC values in the literature, which range from 0.962 to 1.0 [30], and significantly surpass earlier benchmarks of 0.85 [31].

This variability in AUC outcomes can be attributed to differences in the data used (such as sample size, data quality, and specific population characteristics) and the analytical methods employed (including the types of features utilized and data preprocessing techniques). Each study may employ different datasets and methodologies, which can significantly affect the resulting AUC values.

Previous research by García-Pretelt et al. [32] applies machine learning to classify individuals at risk for Alzheimer's disease using resting-state EEG, achieving an impressive accuracy of up to 83% using spatial filters obtained from a gICA approach. Additionally, the study by Francisco Gerson A de Meneses et al. [33] provides insights by using convolutional neural networks (CNNs) to classify neurological diseases based on cortical topographies. The remarkable performance of SqueezeNet, with accuracies of 88.89% for Parkinson's disease, 75.70% for depression, and 72.10% for bipolar disorder, highlights the potential of advanced machine learning techniques in the classification of neurological conditions, which complements our study's focus on ICA configurations.

Caroline L. Alves et al. [34] focuses primarily on harmonizing EEG data from different cohorts, with accuracies of 98 and 99% using CNNs in Parkinson's disease. On the other hand, the present study takes a broader approach, integrating and analyzing data from different sources to improve Alzheimer's risk classification. This strategy has allowed us to gain a more complete and accurate understanding of the impact of the PSEN1-E280A mutation in different carrier contexts. This underscores the importance of considering data diversity in AD research to ensure model robustness and generalizability.

The results of Gerson et al. [33] serve as a reference point, demonstrating the successful application of similar algorithms with EEG data in various diseases, thus reinforcing the robustness of our methods. This, together with other papers [35], highlights the importance of our findings in the clinical context and underlines the implications for future research and clinical applications. However, our study goes further by providing a more comprehensive understanding of how integrating data from different cohorts can improve the accuracy and generalizability of EEG-based AD detection models. This underscores the need to continue to explore innovative and collaborative approaches to address challenges in Alzheimer's diagnosis and treatment.

On the other hand, the components obtained from the reproducibility approach reported in the study by Ochoa-Gómez et al. [17] provide a variety of metrics for classification, in line with the proposal of Prado et al. [36]. This study highlights the need for systematic harmonization in EEG connectivity studies, addressing critical sources of variability and suggesting a composite metric strategy to improve replicability in multicenter studies.

While our study is promising, it's important to acknowledge its limitations. Despite using a comprehensive dataset, we must question its representativeness and potential biases, especially with similar cohorts such as UdeA1 and UdeA2. Generalizing our findings beyond these specific cohorts may be challenging, highlighting the need for validation in diverse populations and increasing cohort diversity [37]. In addition, despite the use of advanced machine learning techniques such as EEG-based classification, the complexity of the models may hinder full interpretation. Finally, potential biases, measurement errors, and underlying assumptions in our analysis models warrant careful consideration and further discussion, particularly regarding model selection and the preference for SVM in the literature.

## Conclusions

Our study represents a significant step forward in the field of computational neuroscience, particularly in the area of Alzheimer's disease detection using EEG data. By harmonizing data from multiple cohorts and applying advanced machine learning techniques, we have developed a robust model capable of accurately discriminating between healthy non-carrier subjects (HC) and E280A mutation Alzheimer's disease carriers without clinically detectable cognitive impairment (ACr).

Our results highlight the importance of integrating data from multiple sources to improve the generalizability and applicability of disease detection models. The successful application of techniques such as propensity score matching (PSM) and group independent component analysis (gICA) highlights the effectiveness of comprehensive approaches in improving model accuracy and reliability.

While our study presents promising results, it is imperative to acknowledge its limitations. The representativeness of our dataset and potential biases inherent in our sample must be carefully considered. Furthermore, the complexity of our machine learning model requires thorough sensitivity analyses and cross-validation to ensure its stability and robustness.

In conclusion, our work contributes valuable insights to the burgeoning field of computational neuroscience, provides a refined understanding of the impact of the PSEN1-E280A mutation, and paves the way for more accurate and early detection methods for Alzheimer's disease. We believe that our findings have significant clinical and research implications and represent a critical step in addressing the challenges posed by neurodegenerative diseases in our aging population.

## Supporting information

**S1 Fig. Quality-control metrics applied across the EEG pre-processing pipeline, including early-stage preprocessing (PREP), wavelet–ICA artifact removal, and noisy time rejection.**
(TIF)

**S1 File. This file contains all supplementary tables generated during the EEG data integration and harmonization procedures.** Tables labeled sovaharmony correspond to the pre-enrichment approach, in which a single harmonization step is applied prior to downstream analyses. Tables labeled neuroHarmonize correspond to the sample enrichment approach, where harmonization is performed after data integration under different group ratio configurations. The archive also includes the table features_p_bonferroni, which reports the original p-values and Bonferroni-corrected p-values for all features retained in the sample enrichment pipeline and used as input to the machine learning models for each group ratio configuration.
(ZIP)

## Ackowledgements

The authors would like to thank the volunteers who participated in the experiments.

## Author contributions

**Conceptualization:** Veronica Henao Isaza, John Fredy Ochoa Gómez.

**Data curation:** Veronica Henao Isaza.

**Formal analysis:** Veronica Henao Isaza.

**Investigation:** Veronica Henao Isaza.

**Methodology:** Veronica Henao Isaza.

**Resources:** David Aguillon, Francisco Lopera.

**Software:** Veronica Henao Isaza.

**Supervision:** David Aguillon, Carlos Andrés Tobón Quintero, John Fredy Ochoa Gómez.

**Validation:** Veronica Henao Isaza.

**Visualization:** Veronica Henao Isaza.

**Writing – original draft:** Veronica Henao Isaza.

**Writing – review & editing:** Veronica Henao Isaza.

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
