## [Decision Letter · Decision Letter 0]

20 Nov 2025

Dear Dr. Ochoa Gómez,

**I agree with the reviewers about the need for further revisions of the manuscript.**

We look forward to receiving your revised manuscript.

Kind regards,

Diego A. Forero, MD; PhD

Academic Editor

PLOS ONE

**Journal Requirements:**

1. When submitting your revision, we need you to address these additional requirements. Please ensure that your manuscript meets PLOS ONE's style requirements, including those for file naming. The PLOS ONE style templates can be found at https://journals.plos.org/plosone/s/file?id=wjVg/PLOSOne_formatting_sample_main_body.pdf and https://journals.plos.org/plosone/s/file?id=ba62/PLOSOne_formatting_sample_title_authors_affiliations.pdf 2. Thank you for stating the following in the Acknowledgments Section of your manuscript: The authors would like to acknowledge the support of the Comité para el Desarrollo de la Investigación (CODI) at the Universidad de Antioquia, through the following research projects:• “Cambios en los patrones del electroencefalograma cuantitativo (reactividad alfa, theta y su índice) en reposo y tareas de memoria, en el seguimiento longitudinal de pacientes con riesgo genético para Enfermedad de Alzheimer Temprano”, project code 2017-16371.• “Evaluación de la actividad eléctrica cerebral en sujetos con deterioro cognitivo mediante electroencefalografía portable y modelos de Machine Learning”, project code PRG 2022-53407. We note that you have provided funding information that is not currently declared in your Funding Statement. However, funding information should not appear in the Acknowledgments section or other areas of your manuscript. We will only publish funding information present in the Funding Statement section of the online submission form. Please remove any funding-related text from the manuscript and let us know how you would like to update your Funding Statement. Currently, your Funding Statement reads as follows: The author(s) received no specific funding for this work.  Please include your amended statements within your cover letter; we will change the online submission form on your behalf. 3. Thank you for uploading your study's underlying data set. Unfortunately, the repository you have noted in your Data Availability statement does not qualify as an acceptable data repository according to PLOS's standards. At this time, please upload the minimal data set necessary to replicate your study's findings to a stable, public repository (such as figshare or Dryad) and provide us with the relevant URLs, DOIs, or accession numbers that may be used to access these data. For a list of recommended repositories and additional information on PLOS standards for data deposition, please see https://journals.plos.org/plosone/s/recommended-repositories. 4. PLOS requires an ORCID iD for the corresponding author in Editorial Manager on papers submitted after December 6th, 2016. Please ensure that you have an ORCID iD and that it is validated in Editorial Manager. To do this, go to ‘Update my Information’ (in the upper left-hand corner of the main menu), and click on the Fetch/Validate link next to the ORCID field. This will take you to the ORCID site and allow you to create a new iD or authenticate a pre-existing iD in Editorial Manager. 5. If the reviewer comments include a recommendation to cite specific previously published works, please review and evaluate these publications to determine whether they are relevant and should be cited. There is no requirement to cite these works unless the editor has indicated otherwise. 

Reviewers' comments:

**Comments to the Author**

1. Is the manuscript technically sound, and do the data support the conclusions?

Reviewer #1: Yes

Reviewer #2: Partly

2. Has the statistical analysis been performed appropriately and rigorously?

Reviewer #1: Yes

Reviewer #2: Yes

3. Have the authors made all data underlying the findings in their manuscript fully available?

Reviewer #1: Yes

Reviewer #2: Yes

4. Is the manuscript presented in an intelligible fashion and written in standard English?

Reviewer #1: Yes

Reviewer #2: Yes

**Reviewer #1:**  The manuscript presents a well-structured and methodologically robust framework for augmenting EEG datasets in Alzheimer’s risk classification through harmonization and propensity score matching. The integration of multi-site datasets and open-science practices adds considerable value. However, the study would benefit from clearer statistical testing, additional model benchmarking, and refinement of ethical statements.

１）Although the effect sizes (Cohen’s d) are presented for group differences (ACr vs. HC), no p-values or statistical significance tests (e.g., t-test, Mann–Whitney U test, ANOVA) are reported. Because effect size alone cannot determine the direction or probabilistic significance of the difference, statistical tests should be provided alongside it.

２）No 95% confidence intervals (CIs) are reported for key performance metrics such as AUC, accuracy, and recall. To demonstrate the stability of the model’s performance, the distribution and confidence intervals across cross-validation folds should be reported.

３）Although 100 features were selected from an initial set of 967, no correction for multiple comparisons (e.g., Bonferroni, FDR) is described when performing statistical tests across multiple features. Such correction is necessary to prevent false positives (Type I errors).

**Reviewer #2:**  This study discusses an EEG-based data augmentation framework by integrating advanced signal processing, component-level feature extraction, data harmonization (neuroHarmonize), and Propensity Score Matching (PSM). The proposed framework addresses sample size and variability constraints in EEG-based AD risk classification. Significance: Harmonization and statistical balancing provide a replicable strategy for multicenter EEG studies targeting early AD detection. Essentially, these findings can improve classification accuracy with limited data. However, several important issues require clarification:

1. We know that ECG signals tend to be random and vary widely between individuals. How do we address this issue?

2. The augmentation method needs to be explained in detail, referring to the manuscript title.

3. Quality testing of the augmented data also needs to be conducted.

4. How the recordings were made, including duration, stimulus, and patient condition.

5. Criteria for inclusion and exclusion of subjects in private datasets should be presented in diagram form.

6. Statistical tests should also be presented in detailed graphical form.

**Do you want your identity to be public for this peer review?** For information about this choice, including consent withdrawal, please see our Privacy Policy

Reviewer #1: No

Reviewer #2: No

---

## [Author Response · Author response to Decision Letter 1]

7 Jan 2026

Reviewer Number 1

The manuscript presents a well-structured and methodologically robust framework for augmenting EEG datasets in Alzheimer’s risk classification through harmonization and propensity score matching. The integration of multi-site datasets and open-science practices adds considerable value. However, the study would benefit from clearer statistical testing, additional model benchmarking, and refinement of ethical statements.

1. Although the effect sizes (Cohen’s d) are presented for group differences (ACr vs. HC), no p-values or statistical significance tests (e.g., t-test, Mann–Whitney U test, ANOVA) are reported. Because effect size alone cannot determine the direction or probabilistic significance of the difference, statistical tests should be provided alongside it.

We thank the reviewer for this comment. In response, we have now included statistical significance tests alongside the reported effect sizes. Specifically, we applied the Mann–Whitney U test, a non-parametric test, for all features to assess differences between healthy non-carrier subjects (HC) and asymptomatic E280A mutation carriers (ACr). The Mann–Whitney U test was chosen because it does not assume normality of the data, which justifies not using a t-test or other parametric tests.

The resulting p-values are reported on the distribution plots together with Cohen’s d, allowing both the magnitude and statistical significance of the differences to be evaluated. P-values are formatted as follows: p < 0.0001, p < 0.001, p < 0.01, p < 0.05, and p = n.s. if not significant.

All figures, including supplementary figures, now present effect sizes and their corresponding Mann–Whitney U test significance, providing a complete statistical summary.

2. No 95% confidence intervals (CIs) are reported for key performance metrics such as AUC, accuracy, and recall. To demonstrate the stability of the model’s performance, the distribution and confidence intervals across cross-validation folds should be reported.

Thank you for the comment. We have now updated Table 3 to include the 95% confidence intervals (CIs) for all key performance metrics (Accuracy, AUC, Recall, and Precision) across cross-validation folds. The table reports the mean ± standard deviation over 10 folds, along with the lower and upper bounds of the 95% CI.

Table 3. Classification performance metrics across matching ratios.

3. Although 100 features were selected from an initial set of 967, no correction for multiple comparisons (e.g., Bonferroni, FDR) is described when performing statistical tests across multiple features. Such correction is necessary to prevent false positives (Type I errors).

We thank the reviewer for highlighting the importance of controlling for multiple comparisons. To address this concern, we applied a Bonferroni correction across all tested features. Specifically, group differences for each numerical feature were assessed using a two-sided non-parametric Mann–Whitney U test, and the resulting p-values were adjusted according to the total number of features tested.

In parallel, Cohen’s d effect sizes were computed for each feature using the difference in group means normalized by the pooled standard deviation, providing a complementary measure of the magnitude of group differences independent of statistical significance.

Only features with Bonferroni-corrected p-values below the predefined significance threshold (p_Bonferroni < 0.05) were retained for subsequent machine learning analyses. This strategy ensures that the models are trained exclusively on features exhibiting statistically robust group differences after strict correction for multiple testing. All Bonferroni-adjusted p-values and corresponding effect sizes are reported in the Supplementary Material.

Reviewer Number 2

This study discusses an EEG-based data augmentation framework by integrating advanced signal processing, component-level feature extraction, data harmonization (neuroHarmonize), and Propensity Score Matching (PSM). The proposed framework addresses sample size and variability constraints in EEG-based AD risk classification. Significance: Harmonization and statistical balancing provide a replicable strategy for multicenter EEG studies targeting early AD detection. Essentially, these findings can improve classification accuracy with limited data. However, several important issues require clarification:

1. We know that ECG signals tend to be random and vary widely between individuals. How do we address this issue?

We thank the reviewer for raising this important point regarding the variability and apparent randomness of EEG signals across individuals. This concern is explicitly addressed in the manuscript through the use of a standardized and robust methodological framework.

As described in the Materials and Methods section, inter-subject variability is mitigated by applying a comprehensive EEG preprocessing pipeline, including standardized artifact removal (PREP, ICA, and wavelet–ICA), signal normalization to account for anatomical differences (e.g., scalp, skull, and hair), extraction of physiologically meaningful features (spectral, entropy, and connectivity measures), and multi-site harmonization using ComBat-based methods (neuroHarmonize). These steps allow us to reduce non-neuronal sources of variability while preserving biologically relevant patterns associated with the condition of interest.

To further improve clarity and explicitly link these methodological choices to the issue raised by the reviewer, we have added the following sentence at the beginning of the EEG data pre-processing section:

“Given the well-known inter-subject variability and stochastic nature of EEG signals, a standardized preprocessing, normalization, and harmonization strategy was applied to minimize non-neuronal variability while preserving biologically meaningful patterns.”

Additionally, part of the observed variability can be influenced by sample size imbalance between groups. This was addressed by implementing propensity score matching at different HC:ACr ratios, allowing us to evaluate the stability of the results under varying sample size conditions.

We believe that this addition clarifies how the proposed methodology directly addresses the inherent variability of EEG signals across individuals.

To illustrate these points, we have included several example figures generated from our data, showing the distribution of EEG metrics across components, bands, and datasets (pre-enrichment vs. sample enrichment). These figures were created using a rigorous plotting strategy that visualizes inter-subject variability while highlighting the effect of harmonization. Specifically, boxplots represent the distribution of metric values across participants, stratified by database and dataset, while overlaid points indicate individual subject data.

2. The augmentation method needs to be explained in detail, referring to the manuscript title.

We thank the reviewer for their insightful comment regarding the augmentation method and its relationship to the manuscript title. We acknowledge that the term “augmentation” may be interpreted as referring to synthetic data generation or signal-level data augmentation techniques.

To avoid ambiguity and to more accurately reflect the methodology employed in this study, we have revised the manuscript by replacing the term “augmentation” with “sample enrichment” throughout the text, including the title, abstract, and results sections. In this work, sample enrichment refers to a cohort-level strategy based on the integration of multi-site EEG datasets, feature harmonization across recording sites using ComBat-based methods (neuroHarmonize), and controlled group balancing through Propensity Score Matching at different HC:ACr ratios. No synthetic EEG signals or artificial data generation procedures were applied.

We believe that this terminology more precisely captures the nature of the proposed framework and improves the clarity and interpretability of the manuscript in line with the reviewer’s comment.

3. Quality testing of the augmented data also needs to be conducted.

We thank you for this valuable comment regarding quality testing of the enriched dataset. We have clarified that quantitative quality-control metrics were applied at multiple stages of the EEG pre-processing pipeline to assess data reliability. A schematic summary of these metrics is provided in the Supplementary Materials (Supplementary Figure S1), ensuring transparent evaluation of data quality prior to feature extraction. Importantly, after applying sample enrichment through Propensity Score Matching at 2:1, 5:1, and 10:1 ratios, we verified that these quality-control metrics remained within acceptable ranges. This confirms that the enrichment process did not introduce additional noise or artifacts, maintaining the reliability of the dataset for subsequent feature extraction and classification.

4. How the recordings were made, including duration, stimulus, and patient condition.

We thank the reviewer for the comment regarding the clarity of the EEG recording details. To address this, we have revised the “Subjects and EEG Acquisition” section to provide a clearer and more comprehensive description of all recordings. The revised section now explicitly includes:

• The resting-state condition of all recordings.

• Duration of recordings for each database.

• Patient condition, specifying that ACr participants did not show clinically detectable cognitive impairment.

• Information on stimulus (none), EEG systems used, and electrode placement.

5. Criteria for inclusion and exclusion of subjects in private datasets should be presented in diagram form.

Thank you for this valuable suggestion. We have now explicitly incorporated the inclusion and exclusion criteria for subjects in private datasets and presented them in diagram form. A new schematic workflow (Fig. 1) has been added to the Subjects and EEG Acquisition section, illustrating the dataset and subject selection process prior to EEG pre-processing. This diagram summarizes the criteria applied to include resting-state EEG datasets and to exclude datasets based on acquisition characteristics, electrode count, preprocessing status, and data availability and consent.

6. Statistical tests should also be presented in detailed graphical form.

We thank the reviewer for the suggestion regarding the graphical presentation of statistical tests. In response, all figures, including supplementary figures, have been updated to provide a detailed statistical summary for each feature. Specifically:

• Effect sizes (Cohen’s d) are reported alongside p-values obtained from non-parametric Mann–Whitney U tests, allowing both the magnitude and statistical significance of group differences (ACr vs. HC) to be evaluated.

• P-values are presented on the plots using the following format: p < 0.0001, p < 0.001, p < 0.01, p < 0.05, and p = n.s. for non-significant comparisons.

• To account for multiple comparisons across the selected features, Bonferroni correction was applied, and only statistically robust features are highlighted in the figures.

• All plots now clearly indicate both effect size and statistical significance, providing a comprehensive and interpretable visualization of group differences.

---

## [Decision Letter · Decision Letter 1]

11 Feb 2026

Comprehensive methodology for sample enrichment in EEG biomarker studies for Alzheimer’s risk classification

PONE-D-25-34782R1

Dear Dr. Ochoa Gómez,

We’re pleased to inform you that your manuscript has been judged scientifically suitable for publication and will be formally accepted for publication once it meets all outstanding technical requirements.

Kind regards,

Diego A. Forero, MD; PhD

Academic Editor

PLOS One

Additional Editor Comments (optional):

Reviewers' comments:

Reviewer's Responses to Questions

**Comments to the Author**

Reviewer #1: All comments have been addressed

Reviewer #2: All comments have been addressed

2. Is the manuscript technically sound, and do the data support the conclusions?

Reviewer #1: Yes

Reviewer #2: Yes

3. Has the statistical analysis been performed appropriately and rigorously?

Reviewer #1: Yes

Reviewer #2: Yes

4. Have the authors made all data underlying the findings in their manuscript fully available?

Reviewer #1: Yes

Reviewer #2: (No Response)

5. Is the manuscript presented in an intelligible fashion and written in standard English?

Reviewer #1: (No Response)

Reviewer #2: Yes

Reviewer #1: The authors have thoroughly addressed all comments raised by Reviewer 1. In particular, the revised manuscript now includes appropriate statistical significance testing alongside effect size reporting, with the use of non-parametric Mann–Whitney U tests and correction for multiple comparisons. The addition of 95% confidence intervals for key performance metrics across cross-validation folds substantially improves the statistical rigor and transparency of the results.

Furthermore, the clarification of the feature selection strategy, the explicit application of Bonferroni correction, and the improved presentation of results in both tables and figures strengthen the methodological robustness of the study. Ethical considerations and data access statements have also been refined and are now clearly documented.

Overall, the manuscript presents a well-validated and clearly described framework for EEG-based sample enrichment in Alzheimer’s disease risk classification. The revisions have significantly improved clarity, reproducibility, and statistical soundness.

Reviewer #2: The authors have made substantial revisions in response to the previous comments. Notably, they have provided a clear explanation regarding inter- and intra-individual EEG bias issues by applying a comprehensive EEG preprocessing pipeline, including standardized artifact removal (PREP, ICA, and wavelet–ICA), signal normalization, and related procedures. The inclusion and exclusion criteria have been clearly presented in a figure. Furthermore, the details of data acquisition and the statistical analyses have been described adequately.

**Do you want your identity to be public for this peer review?** For information about this choice, including consent withdrawal, please see our Privacy Policy

Reviewer #1: No

Reviewer #2: No

---

## [Editor Report · Acceptance letter]

PONE-D-25-34782R1

PLOS One

Dear Dr. Ochoa Gómez,

I'm pleased to inform you that your manuscript has been deemed suitable for publication in PLOS One. Congratulations! Your manuscript is now being handed over to our production team.

Kind regards,

on behalf of

Dr. Diego A. Forero

Academic Editor

PLOS One